# Extent and causes of the collapse in the registration of innovative medications in Lebanon: A mixed-methods analysis

Wadih Mina[1], Hanadi Nahas[2], Maria Rita Lteif[2], Rita Karam[3], Fadi El-Jardali[4], Louis Garrison[5], Soumana C. Nasser[1]*

1 School of Pharmacy, Lebanese American University, Beirut, Lebanon, 2 AccessCore, Beirut, Lebanon, 3 Faculty of Sciences and Medical Sciences, Lebanese University, Hadath, Lebanon, 4 Health Management and Policy Department, American University of Beirut, Beirut, Lebanon, 5 The Comparative Health Outcomes, Policy, and Economics (CHOICE) Institute Department of Pharmacy, University of Washington, Seattle, Washington, United States of America

☯ These authors contributed equally to this work.
* soumana.nasser@lau.edu.lb

## Abstract

### Objectives

Delays in innovative drug registration across countries, or the "drug approval lag", can cause inequities in treatment access, thereby worsening patient outcomes. Registration delays hinder the first step in making treatments available, and are often linked to regulatory inefficiencies, constrained healthcare financing, and fragmented decision-making. Since late 2019, Lebanon's health system has faced overlapping socioeconomic and political crises, yet their impact on innovative medication registration remains undocumented. This study aimed to measure this impact by comparing Lebanon's drug approval lag before (2014–2019) and after (2020–2024) the crisis and to develop an interpretive framework exploring the rationale behind an informal policy to delay innovative medication registration.

### Methods

A mixed-methods approach was adopted. Innovative medications approved by the Food and Drug Administration (FDA) or the European Medicines Agency (EMA) between 2014 and 2024 were included and compared to local registration timelines in Lebanon. In-depth interviews with policymakers, industry leaders, and healthcare providers informed the interpretive framework.

### Results

Findings revealed a dramatic fall in innovative medication registrations post-crisis over these two periods. The proportion of FDA-approved innovative medications registered in Lebanon dropped from 43.6% to 0% and EMA-approved medications

**Data availability statement:** All relevant data are within the manuscript and its Supporting Information files.

**Funding:** The author(s) received no specific funding for this work.

**Competing interests:** The authors have declared that no competing interests exist.

dropped from 59.4% to 0%. Pre-crisis, average registration time was under two years; post-crisis, delays are estimated to exceed four years. Our interpretive framework suggests the intermediate effects of delaying innovative medication registration are mainly to control costs and reduce reimbursement pressures on the Ministry of Public Health. However, key stakeholders believe the resulting negative consequences, such as reduced access to life-saving treatments and greater dependence on parallel market importation, outweigh the short-term benefits.

## Conclusion

Health systems are complex adaptive systems, where policies affecting innovative drug approvals may not only delay access, but also trigger unintended consequences. In Lebanon, the registration of innovative medications should remain independent from reimbursement decisions and grounded in evidence-informed health policies.

---

### Introduction

A well-functioning health system, ideally guided by evidence-informed policies, is crucial for improving patient outcomes, addressing public health needs, and ensuring timely access to innovative health technologies [1,2]. Access to such innovations, through their availability, accessibility, acceptability, and affordability is essential for effective disease management and must be regarded as a global health priority [3]. The availability of medications, often reflected in healthcare systems through regulatory approvals, is recognized as the first step toward achieving accessibility [4,5].

From a global health perspective, registration delays across countries lead to unequal treatment access, ultimately compromising health outcomes [6]. This delay, identified as the *drug approval lag*, can be categorized into: [1] the *drug availability gap*, which is the difference in the *number* of available drugs across countries, and [2] the *time-to-approval lag*, which is the *time* difference for a new drug's approval from its global launch [7,8]. Causes of the drug approval lag are complex and interacting, including regulatory delays, health and pricing policies, insufficient human regulatory capital, and non-attractive markets to pharmaceutical companies [9–12]. However, such delays remain context-specific and are shaped by the structure and priorities of individual health systems.

There are significant cross-country differences in the average time required to launch new medicines [13,14]. The Middle East and North Africa (MENA) region experiences an average delay of 42 months in launching new medicines after their first global launch [15]. In comparison, G20 countries report delays between 8–81 months and the European Union (EU) reports an average delay of 24-months [15]. Between 2012 and 2021, only 17% of these new medicines were introduced in MENA markets, and just 2% became available within one year of their global launch [15]. As of 2021, Lebanon, which ranked among the top 5 countries in the MENA region to have 17% of globally launched new medicines, experienced the sharpest decline

[15]. Newly launched medicines dropped from 29% (2008–2017) to 20% (2012–2021) in Lebanon, compared to drops from 25% to 21% in Egypt and 13% to 11% in Jordan [15]. These high-level findings indicate significant disparities in the availability and timely launch of innovative medications across countries, emphasizing the need for targeted policies to reduce access delays.

To our knowledge, there are only three published studies in the Middle East that evaluated the drug lag, two of which are from high-income countries and one from an upper middle income country. A comparative study on oncology medication approvals found that the Kingdom of Saudi Arabia (KSA) approved 52.5% of FDA-approved oncology medications, with a median time-to-approval lag of 30 months [16]. Evidence from the United Arab Emirates (UAE) further indicated that it outpaced KSA in launching nearly all new products [17]. From a different context, the national regulatory authority in Iran has been influenced by political, social, and economic factors, with average drug approval lags of 65.2 months compared to the FDA and 70.3 months compared to the EMA [18]. These findings indicate a need for deeper research on the region's diverse health system structures.

Lebanon, one the smallest countries in the Middle East, has been through multifaceted crises since October 2019, including: an economic crisis coupled by the depreciation of the national currency by more than 95%; the COVID-19 pandemic; the 2020 Beirut port blast that devastated the country's capital; and the recent 2024 war [19]. The economic crisis led to the reclassification of Lebanon from an upper middle income country to a lower middle income country by the World Bank [20]. These shocks severely disrupted the pharmaceutical sector, regulated by the Ministry of Public Health (MOPH) through policies on importation, regulatory approval, and pricing prior to market authorization [21]. Lebanon's heavy reliance on imports (78.5% of medicines) combined with depleting foreign reserves lifted medication subsidies and left the Lebanese Syndicate of Drug Importers $400 million in debt to international suppliers [21,22]. This contributed to nationwide medication shortages and made healthcare, including treatment for diabetes, cancer, and multiple sclerosis, increasingly inaccessible and unaffordable [23]. The national social security fund which once covered 85% of healthcare expenses, dropped to 10% post crisis, leading to catastrophic healthcare expenditure by 2022 [24]. The MOPH, as the payer of last resort for those without any other form of coverage, also faced significant constraints post economic collapse [25]. Compounding these challenges, the 2024 war is estimated to have reduced Lebanon's real GDP growth by at least 6.6% [26], damaged numerous healthcare facilities, and further strained the health workforce [27].

Once a regional leader in adopting new medicines, with its health system ranked 23rd globally for efficiency, Lebanon's health sector has been profoundly affected by recent crises, resulting in heightened uncertainty regarding access to innovative therapies [28]. To our knowledge, no prior research has quantified Lebanon's drug approval lag over time or explored the policy rationale behind registration collapse during crisis. Empirical evidence is needed to inform future regulatory reform that will benefit the government, healthcare system, and population. Hence, the two objectives for this research study are: [1] to retrospectively and quantitatively measure the drug approval lag in Lebanon over the past 10 years by comparing local registration dates with FDA and EMA approval dates and [2] to develop an interpretive framework to explore the reasoning behind the identified delay in innovative medication registration.

## Materials and methods

### 2.1 Study design

This study followed a time-series mixed-methods design using both quantitative and qualitative data collection and analysis. The drug approval lag in Lebanon was quantitatively measured over a 10 year period (2014–2024), encompassing a pre-crisis (2014–2019) and post crisis period (2020–2024). Qualitative In-Depth Interviews (IDIs) were subsequently conducted with key stakeholders to develop an interpretive framework aimed at understanding how and why the measured delay in the registration of innovative medications occurred [29]. Using an inductive qualitative approach, participants further identified the resulting negative consequences of this delay, explored alternative solutions, and extracted key lessons learned.

**Ethical approval.** This study received ethical approval from the Institutional Review Board of the Lebanese American University (LAU) (IRB#: LAU.SOP.SN1.29/Jan/2025). Eligible participants were invited to participate by sending the IRB approved consent form to their publicly available email addresses. Written consent was obtained from interviewed participants when conducted in person. For those who requested an online interview, consent was provided via email, electronically signed and sent back prior to the time of the interview.

## Quantitative phase addressing objective 1

**Data collection and choice of regulators.** In Lebanon, the registration of innovative medicines falls under the mandate of the Technical Committee (TC) for Drug Registration at the MOPH. This committee follows a structured evaluation process to ensure the safety, efficacy, and quality of marketed pharmaceuticals [30]. In line with Decree No. 571, the MOPH recognizes the FDA, EMA, and all health authorities that align their decisions with WHO recommendations as official reference authorities. Accordingly, the decisions of the FDA and EMA are routinely considered by the TC in its regulatory judgments, which justifies their inclusion in this comparative analysis [31].

Innovative medication approvals between 2014 and 2024 were obtained from three regulatory bodies: the FDA, EMA, and Lebanese MOPH. All necessary data, including the name of innovative medications, molecule, approval date, and other relevant details, are publicly available on each of the respective regulatory body's website [32–34]. When information was missing or when further information was needed we relied on supplementary data sources such as authorization letters, detailed product information, regulatory databases, the FDA Orange and Purple Books, and the WHO's Anatomical Therapeutic Chemical (ATC) classification system.

**Eligibility criteria. Inclusion criteria.** Only "Novel" medications were included in this study. As per the FDA definition, these are innovative medications that contain active ingredients not previously approved or marketed in the US or EU [32]. These often include new molecular entities, offer additional treatment options for patients, and represent advancements in healthcare.

**Exclusion criteria.** Certain medications were excluded to maintain the focus and timeliness of the study analysis. These include: vaccines, generics, biosimilar products, diagnostic agents/solutions, vitamins or nutrition products, stem cell therapies, chelating agents, electrolyte solutions, over the counter medications, new formulations of previously approved medications, new indications of previously approved medications, and combination products containing non-innovative chemical products.

Vaccines were excluded from this study for two reasons: first, they are excluded from the FDA's list of new molecular entities and new therapeutic biological products [32] and second because the approval and deployment of COVID-19 vaccines during the pandemic followed a unique process under the National Deployment and Vaccination Plan in Lebanon [35]. Including vaccines in the analysis would thus skew and mislead the results.

Generics and biosimilar products were excluded by definition, as they do not meet the definition of novel/innovative medications. The remaining exclusion criteria align with those used in previous studies examining drug approval lag, ensuring methodological consistency and allowing comparisons [16,18,36].

**Statistical analysis. Quantitative outcome measures: the drug availability gap and the time-to-approval lag.** Descriptive analyses were conducted using Excel and SPSS Statistics version 19 to examine drug approval dates, ATC classifications, registration status in Lebanon compared to the FDA and EMA, as well as average and median lag times.

**Drug Availability Gap.** The drug availability gap was calculated by comparing the total number of innovative medications approved by the FDA and/or EMA between 2014–2024 with the number of those subsequently approved by the Lebanese MOPH. This was done by generating a master list of FDA & EMA approved innovative medications, cross-referencing it with local MOPH registration records, and computing the percentage of unregistered medications for each year.

**Time-to-Approval Lag.** The economic and healthcare crisis in Lebanon resulted in a substantial number of medications remaining unregistered during the second observation period (2020–2024). This posed a methodological challenge: a

standard calculation of the time-to-approval lag (i.e., the time delay in months between FDA/EMA approvals and Lebanese registration). Many medications had not been approved in Lebanon by the end of the study period, creating censored data where the event of interest (innovative medication registration) had not yet occurred or was not observed within the timeframe of the study. Hence, we report this skewed distribution in two parts: (1) the distribution of approval delays among medications that were successfully registered and (2) the larger proportion of medications that remained unregistered by the MOPH.

For medications approved before the crisis (2014–2019), time-to-approval lag was calculated by subtracting FDA/EMA and MOPH approval dates. After categorization, SPSS was used to compute descriptive statistics including means, medians, and to summarize delay patterns of the pre-crisis period.

### Qualitative analysis addressing objective 2

**Data collection.** IDIs with key national stakeholders informed the development of the interpretive framework. An interview guide, developed by all study authors, covered perspectives on innovative medication registration in Lebanon, recent policy changes, rationales, outcomes, and recommendations (S1 File).

IDI's, averaging 35 minutes, were conducted by three trained researchers who began with formal introductions to build rapport. IDIs were recorded and manually transcribed verbatim after consent was obtained; otherwise detailed notes were taken. Transcripts were shared with a number of participants to confirm understanding and ensure accuracy of their perspectives. Throughout, researchers maintained reflexivity, making sure no assumptions, subjectivity, or bias influenced the research process [37].

**Eligibility criteria. Inclusion criteria.** Participants were required to have senior-level experience in drug registration, regulatory affairs, market access, public health, health economics, academia, or the pharmaceutical industry in Lebanon. Eligible participants were affiliated with one or more of the following entities:

• The Ministry of Public Health in Lebanon

• Multinational pharmaceutical companies operating in Lebanon

• Pharmaceutical agents, importers, and drug distributors

• Universities and academic institutions

• Healthcare Syndicates

**Sampling and recruitment.** Participants were recruited through purposive sampling based on the study's inclusion criteria, followed by snowball sampling whereby initial interviewees recommended additional experts to participate. The process of recruitment and data collection started from February 27th 2025 till April 29th 2025. Twelve participants agreed to participate after invitation while three refused due to travel commitments or lack of interest. Upon obtaining written consent, interviews were arranged at a time mutually convenient for both the participant and the research team. The interviews were conducted either in person or online depending on participants' preferences.

Participant invitations were sent via email in a gradual and iterative manner until data saturation was achieved [38]. Throughout the data collection phase, the research team continuously assessed for information redundancy and the generation of new ideas. Upon confirming that no new insights were emerging from the interviews, two additional interviews were conducted before concluding the sampling and data collection phases [38].

**Qualitative analysis.** Thematic analysis was conducted following Braun and Clarke's 6-phase guide [39]. Coding was performed manually using a qualitative codebook, which organized the data into codes, subthemes, themes, and supporting quotes. Initial codes were inductively generated through repeated reading of the transcripts, then reviewed, refined, and grouped into broader themes through iterative discussion among three study authors to ensure credibility

and consistency [39]. The major themes identified were used to explain the rationale behind the identified drug approval lag and were incorporated into the interpretive framework. Inductively generated information provided further context on Lebanon's crisis and its connection to the observed delay in innovative medication registration.

## Results

### Drug availability gap

**Drug availability gap calculations.** Between 2014 and 2024, the number of innovative medications approved by the FDA, EMA, and Lebanese MOPH that met our inclusion criteria were as follows: 479 by the FDA, 382 by the EMA, and 139 by the Lebanese MOPH (Fig 1).

The average number of innovative medication approvals per year varied across regulators. The FDA, EMA, and MOPH approved an average of 43.54 (range: 20–57), 34.73 (range: 27–47), and 12.63 (range: 0–25) innovative medications per year.

It is important to note that not all 139 medications approved by the MOPH had received FDA or EMA approval between 2014 and 2024, some were approved before 2014. Therefore, our quantitative analyses include only those medications that received FDA and EMA approval between 2014 and 2024 and were registered by the MOPH during the same period.

Of the 479 innovative medications approved by the FDA between 2014 and 2024, only 80 **(16.7%)** were registered by the Lebanese MOPH during this time period. The remaining 59 that were registered in Lebanon between 2014 and 2024 had received FDA approval prior to 2014.

Of the 382 innovative medications approved by the EMA between 2014 and 2024, only 93 **(24.35%)** were registered by the Lebanese MOPH. The remaining 46 that were registered in Lebanon between 2014–2024 had received EMA approval prior to 2014.

**Drug availability gap in lebanon compared to the FDA per year.** The proportion of registered innovative medications in Lebanon compared to the FDA fell dramatically from 43.6% to 0% between the two periods: 2014–2019 and 2020–2024 (Table 1).

**Drug availability gap in Lebanon compared to the EMA per year.** The proportion of registered innovative medications in Lebanon compared to the EMA fell dramatically from 59.4% to 0% between the two periods: 2014–2019 and 2020–2024 (Table 2).

Tables 1 and 2 highlight the trend of innovative medications approved by the FDA or EMA between 2014 and 2024 and their respective registration in Lebanon. The data reveals a significant decline in innovative registrations after the onset

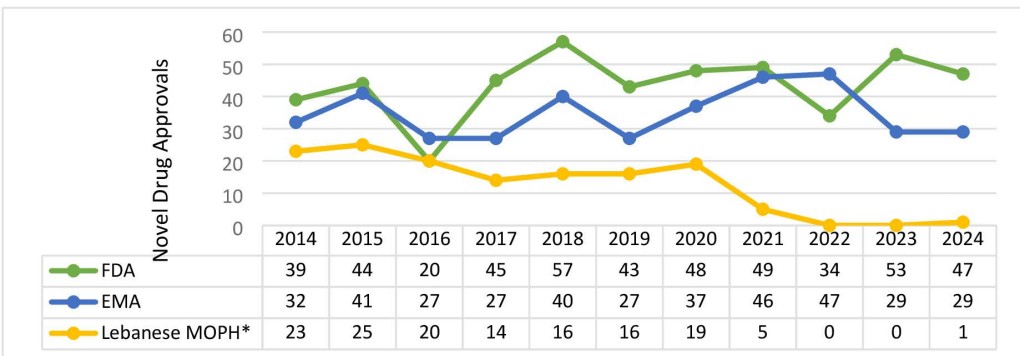

| | 2014 | 2015 | 2016 | 2017 | 2018 | 2019 | 2020 | 2021 | 2022 | 2023 | 2024 |
|---|---|---|---|---|---|---|---|---|---|---|---|
| FDA | 39 | 44 | 20 | 45 | 57 | 43 | 48 | 49 | 34 | 53 | 47 |
| EMA | 32 | 41 | 27 | 27 | 40 | 27 | 37 | 46 | 47 | 29 | 29 |
| Lebanese MOPH* | 23 | 25 | 20 | 14 | 16 | 16 | 19 | 5 | 0 | 0 | 1 |

**Fig 1. Innovative medication approvals by the FDA, EMA, and Lebanese MOPH between 2014-2024.** *Including medications that received either FDA or EMA approval before 2014.

**Table 1.** Drug Availability Gap in Lebanon Compared to the FDA (2014–2024).

| Year | Number of Innovative FDA Approvals | Number that Received Registration in Lebanon | Number that did not Receive Registration in Lebanon | Proportion Registered (%) | Proportion Not Registered (%) |
|---|---|---|---|---|---|
| 2014 | 39 | 17 | 22 | 43.6% | 56.4% |
| 2015 | 44 | 17 | 27 | 38.6% | 61.4% |
| 2016 | 20 | 9 | 11 | 45.0% | 55.0% |
| 2017 | 45 | 17 | 28 | 37.8% | 62.2% |
| 2018 | 57 | 9 | 48 | 15.8% | 84.2% |
| 2019 | 43 | 9 | 34 | 21.0% | 79.0% |
| 2020 | 48 | 1 | 47 | 2.0% | 98.0% |
| 2021 | 49 | 0 | 49 | 0.0% | 100.0% |
| 2022 | 34 | 1 | 33 | 2.0% | 98.0% |
| 2023 | 53 | 0 | 53 | 0.0% | 100.0% |
| 2024 | 47 | 0 | 47 | 0.0% | 100.0% |

**Table 2.** Drug Availability Gap in Lebanon Compared to the EMA (2014–2024).

| Year | Number of Innovative EMA Approvals | Number that Received Registration in Lebanon | Number that did not Receive Registration in Lebanon | Proportion Registered (%) | Proportion Not Registered (%) |
|---|---|---|---|---|---|
| 2014 | 32 | 19 | 13 | 59.4% | 40.6% |
| 2015 | 41 | 16 | 25 | 39% | 61% |
| 2016 | 27 | 14 | 13 | 51.8% | 48.2% |
| 2017 | 27 | 11 | 16 | 40.7% | 59.3% |
| 2018 | 40 | 15 | 25 | 37.5% | 62.5% |
| 2019 | 27 | 7 | 20 | 26% | 74% |
| 2020 | 37 | 8 | 29 | 21.6% | 78.4% |
| 2021 | 46 | 1 | 45 | 2% | 98% |
| 2022 | 47 | 2 | 45 | 4.2% | 95.8% |
| 2023 | 29 | 0 | 29 | 0% | 100% |
| 2024 | 29 | 0 | 29 | 0% | 100% |

of the 2019 crisis. The sharpest decline occurred between 2021 and 2024, during which Lebanon registered almost no innovative medications.

Looking at Table 1, only one FDA-approved innovative medication was registered in Lebanon between 2021 and 2024. This same medication was also EMA approved and presented in Table 2. The remaining two medications approved by the EMA in 2021 and 2022 had previously received FDA approval and were already registered in Lebanon based on that prior approval.

**The drug availability gap before and after the 2019 crisis.** Figs 2 and 3 present the Drug Availability Gap in Lebanon before and after the 2019 crisis compared to the FDA and EMA.

**Comparison of drug availability gap: FDA**

**2014–2019:** During this period, 248 innovative medications were approved by the FDA. Of these, 78 were registered in Lebanon (31.45%), resulting in a drug availability gap of 170 innovative medications.

**2020–2024:** During this period, 231 innovative medications were approved by the FDA. However, only 2 of these medications were registered in Lebanon (0.8%), resulting in a drug availability gap of 229 innovative medications.

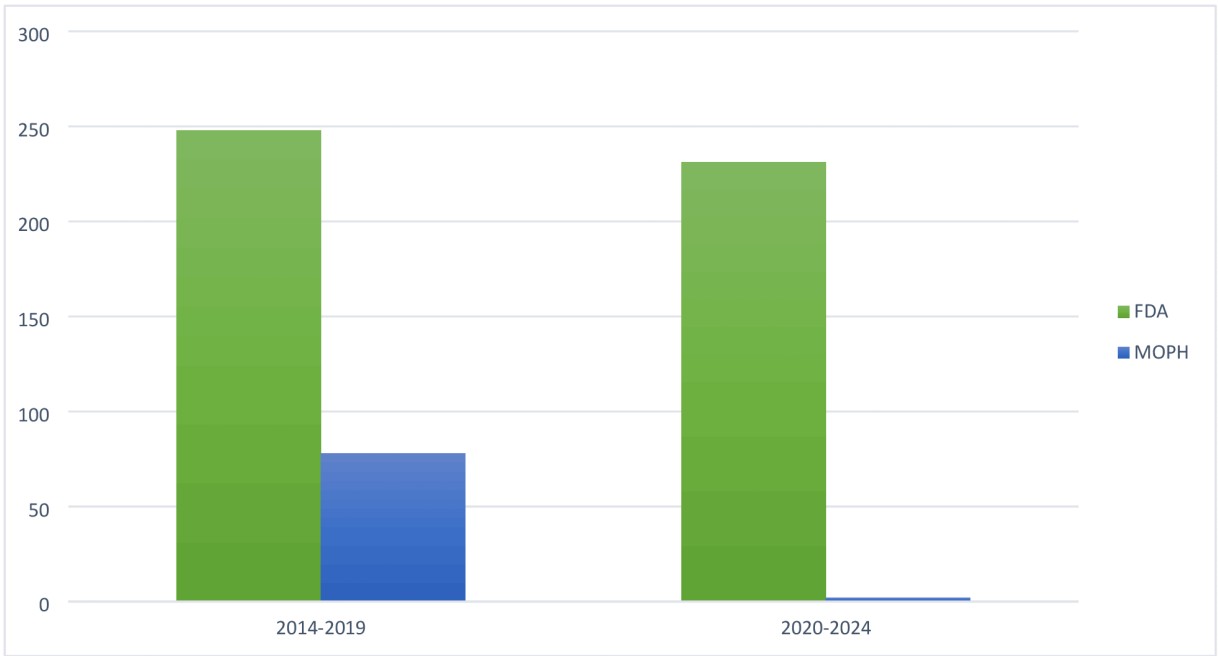

**Fig 2. Drug availability gap in Lebanon before and after the 2019 crisis compared to the FDA.**

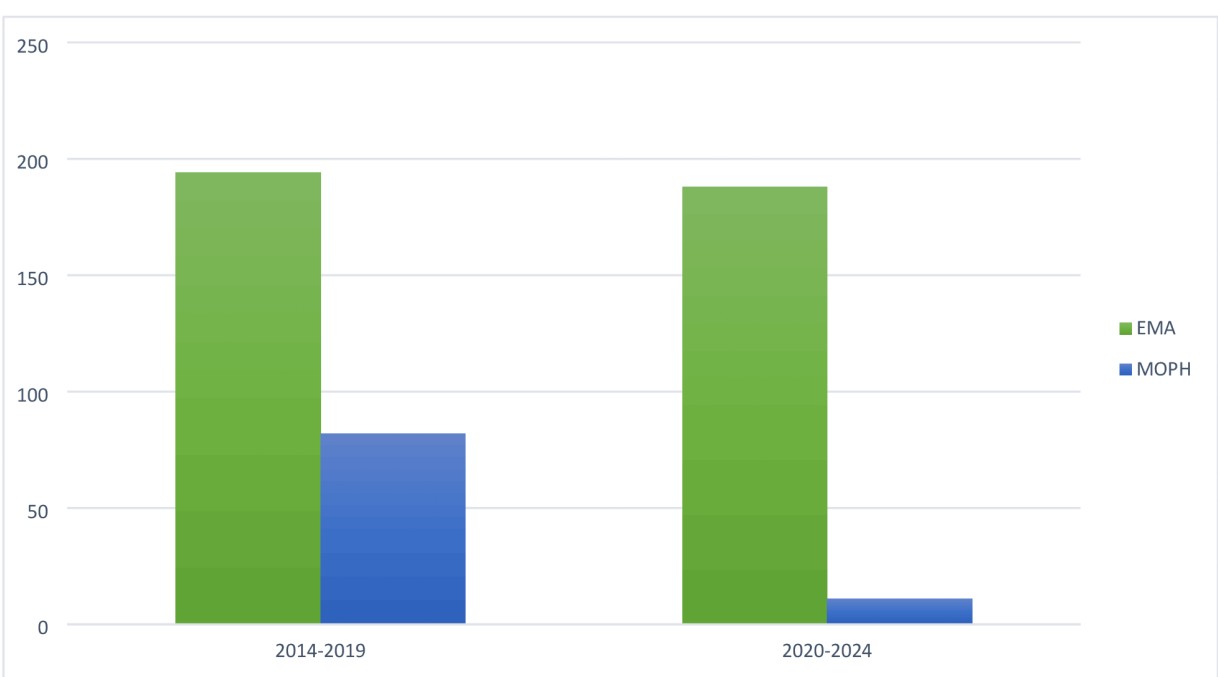

**Fig 3. Drug availability gap in Lebanon before and after the 2019 crisis compared to the EMA.**

## Comparison of drug availability gap: EMA

**2014–2019**: During this period, 194 innovative medications were approved by the EMA. Of these, 82 were registered in Lebanon (42.27%), leading to a drug availability gap of 112 innovative medications.

**2020–2024:** During this period, 188 innovative medications were approved by the EMA. However, only 11 of these medications were registered in Lebanon (5.8%), resulting in a drug availability gap of 177 innovative medications.

**Drug availability gap per therapeutic class.** S1 and S2 Tables compare the drug availability gap in Lebanon relative to both the FDA and EMA across ATC classification. Pre-crisis, antineoplastic and immunomodulating medications (i.e., cancer medications) accounted for the highest number of registrations in Lebanon, reflecting a proportional alignment with FDA and EMA approvals. These results indicate that approximately 50% of antineoplastic and immunomodulating medications approved internationally were also registered in Lebanon pre-crisis. However, post-crisis, this pattern shifted: the few medications that were registered in Lebanon did not follow any clear trend or therapeutic prioritization procedure, suggesting an unstructured approach to medication registration regardless of public health needs or international benchmarks.

## Time-to-approval lag

Since the time-to-approval lag could not be calculated for those unregistered innovative medications after the 2019 crisis (explanation found in section 2.2.3), we proceeded to calculate the time-to-approval lag pre-crisis only (2014–2019) as shown in S3 Table.

Between 2014 and 2019, the mean lag time for innovative medication registration in Lebanon was 19.4 months vs. the FDA and 12.4 months vs. the EMA. The greatest lag time was 70 months compared to both regulators, while the shortest was −4 months compared to the FDA and −41 months compared to the EMA.

Given the large number of medications that remained unregistered post-crisis, the time-to-approval lag in Lebanon likely increased by several years. For instance, over 100 medications did not receive approval during this period, extending their lag by at least 4 years (48 months).

## Qualitative interview results and interpretive framework

A total of 12 in-depth interviews were conducted during the qualitative phase. Six participants were recruited from multinational pharmaceutical companies and importers, four from the public sector (one policymaker, two government officials, and one syndicate president), and two from academic hospitals in Lebanon. This balance between participants' backgrounds was important to make sure the results do not present only a specific perspective of stakeholders, which could have biased the interpretative framework.

A total of 6 themes were generated from the qualitative analysis:

**Stakeholder awareness and framing of the identified drug approval lag.** Participants noted that key stakeholders, including policymakers, industry leaders, healthcare professionals, and syndicates, were aware of the drug approval lag, though awareness varied in timing and extent. From their observations, the delay was not immediately evident during the early stages of the crisis, but over time it became clear to stakeholders that delays in the registration of innovative medications were occurring. Multinational pharmaceutical companies and importers were the first to recognize this issue and raise concerns, while healthcare professionals became aware as unmet clinical needs emerged.

*"Relevant stakeholders, including the Ministry of Public Health, industry leaders, and healthcare professionals, are largely aware of the delays in innovative medication registration. However, the extent of awareness varies."* — *Regulatory Expert, Drug Distributor*

On the other hand, local manufacturers, importers of generic/biosimilar products, community pharmacists, and the general public were perceived to be less aware, as the delay had minimal impact on their operations.

**Root causes of the identified drug approval lag.** Participants identified several root causes contributing to the drug approval lag, including financial challenges, budget constraints, and low purchasing powers, which limited access to costly innovative medications (S1 Fig). This was exacerbated by competing priorities within the MOPH, including the COVID-19 pandemic, the Beirut blast, the central bank's depleting foreign reserves, and the local currency's depreciation. Limited regulatory staff driven by low employee morale, emigration, and strikes also contributed to delayed innovative registration.

> *"The decision was taken based on the MOPH's concern that the healthcare system could not bear the financial burden of registering expensive treatments in the context of a crumbling currency."* — *Regulatory Expert, Drug Distributor*

Two participants noted that Lebanon's diminishing attractiveness as a market for multinational pharmaceutical companies contributed to registration delays, as many were scaling down their operations in the country.

**Rationale behind delaying innovative medication registration.** Fig 4 illustrates the interpretive framework of delaying innovative medication registration as derived from participants' insights. This delay stemmed from an informal policy by key MOPH policymakers to shift registration priorities toward generics and biosimilars. The goal was to avoid introducing high-cost therapies, thereby easing pressure on payers, reducing out-of-expenditures, and conserving limited MOPH resources (public funding for medicine procurement in foreign currency and the strained administrative capacity of the ministry). Budget allocations were redirected to cheaper alternatives, framing this informal policy as a cost-containment strategy. However, many participants perceived this as a reflection of mismanagement during crisis.

Beyond cost savings, policymakers believed it is inequitable for expensive treatments to be accessible only to those with private insurance or the ability to pay. The resulting informal policy was thus aimed at promoting equity and reducing affordability challenges. However, many participants felt that denying access to innovative therapies for the whole population was unjustified, especially when other funding options were possible. Several also noted that the policy was never officially documented or communicated, considering it as "informal".

**Consequences of the identified drug approval lag.** Participants reported consequences of the drug approval lag at three levels: patients, country, and business:

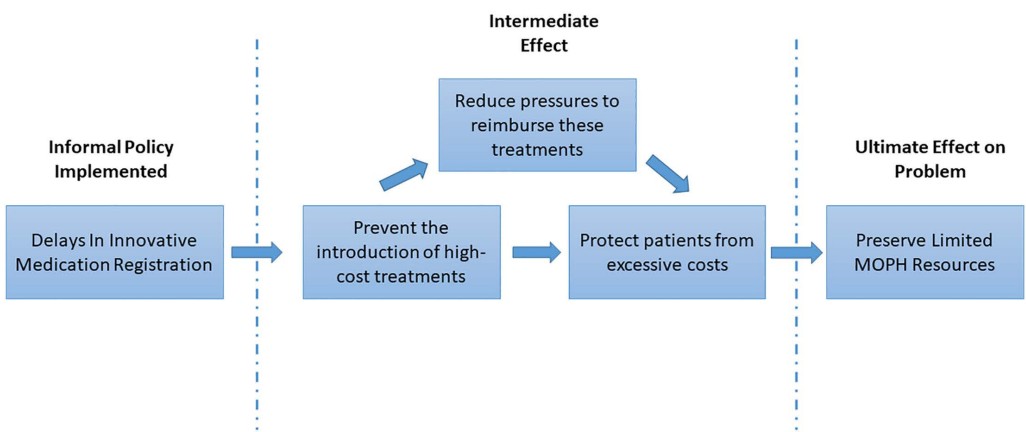

**Fig 4. Interpretive framework illustrating the rationale behind the decision to delay the registration of innovative medications in Lebanon.**

At the patient level, participants emphasized that such drug approval lags delayed access to lifesaving and life-extending treatments, worsening patient outcomes. The absence of innovative therapies were believed to raise costs from treatment failures or relapses and push patients toward parallel markets with unregulated prices and safety risks.

*"Patients are being treated with older molecules that may not be as potent, effective, or, at times, as safe as innovative treatments. Access to life saving medications and life changing treatments stopped, and this for me is the biggest problem that you can have." —General Manager, Multinational Pharma*

At the country level, participants noted that unregistered innovations drove parallel imports of unregulated, counterfeit, and smuggled medications, undermining quality. They also highlighted declining healthcare standards and Lebanon's weakening role as a regional healthcare hub. While most saw this delay as a threat to medical tourism, two disagreed, citing insufficient evidence to support this claim.

*"There isn't medical tourism anymore here in Lebanon. What would the patient come and do, take a medication that is 5 years old? Why would he come? Before 2019 we had many treatment referral programs with embassies where patients from around the world would come to Lebanon to get treated. Now, we only have one program and the rest stopped."— Pharmacist, University Hospital*

In contrast, participants from the MOPH viewed the impact of the drug approval lag as minimal since patients could still import medications through named-patient importation and special requests. Although hospitals were able to import select innovative medications during the crisis, many participants considered this unsustainable, costly for patients, and logistically challenging.

*"As a university hospital I can import medications even if they are not registered, and we were doing that … During the crisis there is one innovative medication that a patient needed and was able to pay for it, we imported it for him through special importation, the medication cost 80,000$ … Although possible, it was definitely not an easy process nor was it the ideal solution."— Pharmacist, University Hospital*

On the business level, multinational pharmaceutical representatives warned that registration delays jeopardize their operations, weaken the economy through reduced business activity, and cause missed clinical trials, financial losses, and office closures.

*"This drug lag is considered a crime against the quality of medications available in the country, the availability of innovative medications, and the presence of scientific offices in Lebanon." — President, Lebanese Syndicate*

**The role and decision to implement Health Technology Assessments (HTA).** Participants were also asked about their opinion behind the ministry's decision to delay innovative medication registration until the establishment of an HTA committee. While most agreed that HTA is important for evidence-based decision-making, they noted that the crisis hindered its implementation. Mentioned challenges to effective HTA implementation include time constraints, limited financial and human resources, the MOPH's dual role as both regulator and payer, and linking HTA decisions to product registration instead of reimbursement.

*"The Lebanese Ministry of Public Health adopted this model [mandating HTA], not merely as a delaying tactic but as a structured approach to reform the drug registration process." — Public Sector Official, Ministry of Public Health*

Three participants (a policymaker, pharmacist, and syndicate president) opposed creating a new HTA committee during the crisis, arguing it would duplicate efforts and drain resources. Instead, they called for implementing the 2022 Lebanese Drug Agency (LDA) law, which mandates a national HTA plan with specific requirements.

**Lessons learned and future projections.** Participants identified two key lessons: avoid delaying innovative medication registration and separate registration from reimbursement within the MOPH. They emphasized that timely registration, even without immediate reimbursement, could have improved patient access to new treatments, preserved Lebanon's reputation as a healthcare hub, and mitigated the aforementioned negative consequences.

Participants stressed that an HTA committee should focus on reimbursement, not registration, to avoid bottlenecks for innovations. Clearing the present registration backlog will also require funding to boost the ministry's review capacity.

Finally, participants emphasized several priority actions, such as strengthening the financial sustainability of Lebanon's healthcare system, updating pharmaceutical legislation (including the implementation of the LDA), and implementing robust institutional frameworks to manage the local budget through evidence-informed decision-making.

## Discussion

This is the first study to assess drug approval delays and access to innovative medications in Lebanon after the crisis, using quantitative descriptive analysis and qualitative key informant interviews. Medication access delays can occur at three levels: (1) from the time of first global approval to local regulatory approval, (2) from local approval to reimbursement, and (3) from reimbursement to proper medication use within the population. In Lebanon, the lag occurred at the first level and was referred to as the "Drug Approval Lag".

The sharpest decline in innovative registration occurred after the 2019 crisis in Lebanon, most notably between 2021 and 2024. The interpretive framework revealed that an informal policy to delay innovative medication registration was intended to mitigate treatment costs on patients, reduce reimbursement pressures on payers, and preserve limited healthcare resources. Although described by most stakeholders as "unwritten" or "informal", this policy aligns with the foundations of a health policy, where a decision, plan, or action is taken that carries significant consequences on society [40]. Severe fiscal constraints affecting both patients and payers, together with Lebanon's declining market attractiveness to multinational pharmaceutical companies, were identified as additional interlinking factors that further delayed the registration of innovative medications. Lebanon's case also demonstrated how limited institutional capacity hinders crisis leadership and undermines the effectiveness and sustainability of responses [41]. Ultimately, the drug approval lag in Lebanon reflects a crisis-driven informal policy response, that is deeply rooted in structural and systemic challenges within the health sector.

Delayed treatment access has major public health implications and is a barrier to equitable healthcare access [8,10]. This is particularly significant for life-saving medications with no alternative treatment options, multi-resistant infections that require innovative therapies, and older medications with poor safety profiles [8]. Additionally, delayed access to oncology medications is linked to losses in person-years of life and quality-adjusted life-years [42]. A country's medical tourism is also impacted by drug approval delays, leading to further economic losses and reduced healthcare competitiveness [43,44]. These implications have not been measured in Lebanon yet, but the growing drug approval lag exacerbates all the aforementioned risks, highlighting the urgent need for corrective action.

Lebanon's experience aligns with evidence linking essential medicine availability to a country's income level [13]. According to recent data, the median time until an essential medicine becomes available in high income countries was 2.7 years, 4.5 years in upper-middle-income countries, 6.9 years in lower-middle-income countries, and 8 years in low-income countries [13]. Pre-crisis, Lebanon saw no major delay in innovative medication availability, registering about 50% of FDA-approved oncology medications and achieving faster registration than KSA (19.75 vs. 34.8 months) [16,17]. However, as Lebanon's income level declined, access to innovative medicines dropped, threatening its regional healthcare standing. Our interpretive framework thus supports evidence from low middle income countries, where delayed product

launches stem from efforts to control healthcare spending, limited human resources, and poor transparency and communication with industry stakeholders [45]. Evidence from other developing countries, such as Zimbabwe and Myanmar, attributes delays in access to innovative medications to limited regulatory capacity, insufficient financial resources, and nontransparent regulatory practices [46,47]. Additionally, pharmaceutical price regulation systems, such as those implemented in Ethiopia, have also been cited as contributing factors to delays in access to essential medicines, drug launches, and market entry [48]. While Lebanon's crises intensified these challenges, the observed delay in innovative medication approvals was largely attributable to financial and policy factors rather than regulatory limitations.

Building on this study's findings, interview insights, and supporting literature, one immediate recommendation and two structural reform measures are proposed to address Lebanon's drug approval lag. The first and most immediate recommendation is to resume the registration of innovative medications in Lebanon and ensure that such processes are never halted or influenced by non–evidence-based decisions, thereby safeguarding patients' access to life-saving therapies [49]. HTA should not be viewed as a bottleneck in the registration of innovative medications, as its primary function is to inform reimbursement decision-making rather than regulatory approval [50]. This is in line with stakeholder recommendations, emphasizing the need to reform the MOPH's dual role as regulator and purchaser in light of the recent political shift. Second, Lebanese policymakers must adopt a transparent, evidence-informed approach to public health decision-making to ensure the success of their policies and stakeholder acceptability [51,52]. This approach may be complex and subject to challenges, including the unpredictability of policy processes, competing political interests, and difficulties in capturing resources, values, or beliefs [53,54]. However, several established frameworks and practical resources are available to facilitate its adoption [51]. These include frameworks endorsed by the WHO (HTA and the compilation of essential medicine lists), and others which emphasize the roles and influence of various interest groups, stakeholders, and coalitions in policymaking (Advocacy Coalition Framework and "3-i" framework). Third, policy makers must recognize that health systems are complex adaptive systems that change and evolve due to multiple interactions and interconnected components [55]. A systems-thinking approach is therefore recommended to guide health policy design and implementation by accounting for system interconnections, anticipating feedback loops to prevent unintended consequences, and modeling potential solutions [55,56]. Additionally, applying a systems perspective requires a multi-stakeholder approach, where users, representatives, and health system leaders are identified, deliberate on the possible effects of proposed policies, conceptualize their effects on the health system, and ultimately adapt and redesign [56]. This reflects the diverse values, ideologies, and political interests of stakeholders, a key factor in complex health systems.

It is important to note limitations to this study. The results were purely descriptive and did not test causal inferences. Future research is needed to explore the relationship between the identified drug approval lag and the resulting negative consequences. The time-to-approval lag could not be calculated post crisis due to the large number of unregistered innovative medications by the end of the observation period. Additionally, the qualitative component is subject to response bias, limited generalizability, and interpretive subjectivity; however, these were mitigated through consistent interviewee reflexivity and qualitative research best practices.

Despite these limitations, this is the first study to systematically examine the drug approval lag in Lebanon, addressing a critical evidence gap amid ongoing crises. The interpretive framework contextualizes delays within broader policy and system dynamics and provides actionable recommendations to support the resumption of innovative medication registration. In 2025, the registration of innovative medications has gradually resumed, making this paper a critical documentation of events during the peak of Lebanon's crisis and a valuable reference to ensure that future policy decisions do not reintroduce drug approval delays—both in Lebanon and in other health systems facing similar instability.

## Conclusion

This study identified a major delay in the registration of innovative medications in Lebanon, emerging after the onset of the 2019 crisis. Causes behind this drug approval lag stem from an informal policy to delay innovative medication registration

in light of financial constraints and limited institutional capacity. Lessons learned and evidence-informed recommendations were provided to ensure the registration of innovative medications in Lebanon resume and remain independent from reimbursement decisions. The Lebanese experience should also be used as a learning for other countries facing instability, highlighting the importance of transparent and evidence-informed health policies.

## Supporting information

**S1 File. Stakeholder Interview Guide.**
(PDF)

**S1 Table. Drug availability gap in Lebanon pre and post crisis compared to the FDA per ATC.** *For products approved in 2024 whose ATC is still pending approval. †Combination of remaining ATC who make a very small proportion of our sample. ‡Therapeutic subgroup under "A – Alimentary tract and metabolism", but it's often reported separately due to its importance.
(DOCX)

**S2 Table. Drug availability gap in Lebanon pre and post crisis compared to the EMA per ATC.** *For products approved in 2024 whose ATC is still pending approval. †Combination of remaining ATC who make a very small proportion of our sample. ‡Therapeutic subgroup under "A – Alimentary tract and metabolism", but it's often reported separately due to its importance.
(DOCX)

**S3 Table. Comparative Time-to-approval lag between FDA, EMA, and Lebanese MOPH between 2014–2019 (pre-crisis).** *The reason the lag time here is negative is because the MOPH had registered this medication based on prior approval by the other respective regulatory body. Meaning, Daclatasvir was approved by the EMA before the FDA, and its registration in Lebanon was based on the EMA's approval.
(DOCX)

**S1 Fig. Visual representation of the main root causes of the identified drug approval lag in Lebanon.**
(TIFF)

## Author contributions

**Conceptualization:** Wadih Mina.

**Formal analysis:** Maria Rita Lteif.

**Investigation:** Wadih Mina, Hanadi Nahas, Maria Rita Lteif, Soumana C. Nasser.

**Methodology:** Wadih Mina, Maria Rita Lteif, Louis Garrison.

**Resources:** Hanadi Nahas.

**Supervision:** Wadih Mina, Hanadi Nahas.

**Validation:** Louis Garrison, Soumana C. Nasser.

**Writing – original draft:** Maria Rita Lteif.

**Writing – review & editing:** Wadih Mina, Hanadi Nahas, Rita Karam, Fadi EL-Jardali, Louis Garrison, Soumana C. Nasser.

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
