## [Decision Letter · Decision Letter 0]

29 Oct 2025

Dear Dr. Nasser,

Thank you for submitting your manuscript to PLOS ONE. After careful consideration, we feel that it has merit but does not fully meet PLOS ONE’s publication criteria as it currently stands. Therefore, we invite you to submit a revised version of the manuscript that addresses the points raised during the review process.

We look forward to receiving your revised manuscript.

Kind regards,

Mabel Aoun, MD, MPH

Academic Editor

PLOS ONE

Journal Requirements:

Reviewers' comments:

Reviewer's Responses to Questions

**Comments to the Author**

1. Is the manuscript technically sound, and do the data support the conclusions?

Reviewer #1: Partly

Reviewer #2: Yes

2. Has the statistical analysis been performed appropriately and rigorously?

Reviewer #1: I Don't Know

Reviewer #2: Yes

3. Have the authors made all data underlying the findings in their manuscript fully available?

Reviewer #1: Yes

Reviewer #2: Yes

4. Is the manuscript presented in an intelligible fashion and written in standard English?

Reviewer #1: Yes

Reviewer #2: Yes

Reviewer #1: My comment concerns section 2.3 Qualitative analysis addressing Objective. Details on sample size calculations were not given (why 15 participants were chosen?What is the rationale behind the choice of participants' number according to their background?). Other aspects should include a summary of study participant characteristics.

My other comment concerns the discussion; In my opinion this paper's second recommendation should come first.

Reviewer #2: The paper addresses a critical and under-researched issue with strong potential policy impact. However, several aspects require deeper methodological rigor, analytical framing, and interpretive depth, particularly in the handling of censored data, qualitative analysis, and the linkage between results and policy recommendations. With some analytical improvements, the manuscript could make a substantial contribution to the literature on regulatory policy and health system resilience in crisis-affected contexts.

Introduction

The knowledge gap is not stated with enough precision. The paper should clearly articulate that no prior research has quantified Lebanon’s drug approval lag over time or explored the policy rationale behind registration collapse in a crisis context.

The conceptual framing of “drug approval lag” is sound but underutilized. The introduction could make stronger and clearer links between the main systemic drivers of such delays, like limited regulatory capacity, declining market attractiveness, and severe fiscal constraints and how these factors specifically apply to Lebanon’s current situation. Doing so would help justify why the study is important and relevant beyond simply describing what happened.

Discussion

1. Shift from description to analysis: the discussion too often repeats the results without moving into deeper interpretation. It should critically analyze why the collapse occurred: for example, how political decision-making, institutional weaknesses, fiscal constraints, and market withdrawal interacted to produce the observed outcome.

2. Comparative and contextual depth: the section lacks sufficient comparative framing. Situating Lebanon’s experience alongside other crisis-affected or resource-constrained countries (LMICs or regional countries) would strengthen external validity and help clarify whether these dynamics are unique or part of broader patterns in drug regulatory behavior.

3. Evidence for consequences: Assertions about patient outcomes, economic losses, and declines in medical tourism are plausible but not directly supported by the study’s data. These should be reframed as hypotheses for future research.

4. Actionable policy recommendations: Recommendations remain high-level and generic. They should be prioritized (e.g., immediate vs. structural reforms) and explicitly linked to the findings. For example, showing how separating registration from reimbursement could directly address the informal policy bottlenecks identified.

5. Integration of policy theory: The discussion would benefit from a stronger theoretical framework, drawing on health policy analysis or crisis governance literature. This would elevate the paper from a descriptive case study to a more analytical contribution with broader policy relevance.

6. Limitations: The treatment of censored data is a major methodological limitation; without appropriate analytical techniques (e.g., survival analysis), time-to-approval estimates may be incomplete or misleading.

The conclusion restates results clearly but should go further in highlighting the key takeaway: that the near halt in drug registration is a systemic issue rooted in institutional capacity, financial constraints, and policy design.

Policy recommendations should focus on two priorities: resuming registration independently of reimbursement decisions and strengthening regulatory governance to prevent future collapses.

Emphasizing how these lessons apply beyond Lebanon, particularly to other crisis-affected health systems would strengthen the paper’s overall contribution.

**Do you want your identity to be public for this peer review?** For information about this choice, including consent withdrawal, please see our Privacy Policy

Reviewer #1: **Yes: ** Myriam Watfa

Reviewer #2: **Yes: ** Jalal Dahham

---

## [Author Response · Author response to Decision Letter 1]

25 Nov 2025

All reviewer and editor comments were addressed in the "Response to Reviewers" document uploaded.

---

## [Editor Report · Decision Letter 1]

11 Dec 2025

Extent and causes of the collapse in the registration of innovative medications in Lebanon: A mixed-methods analysis

PONE-D-25-41837R1

Dear Dr. Nasser,

We’re pleased to inform you that your manuscript has been judged scientifically suitable for publication and will be formally accepted for publication once it meets all outstanding technical requirements.

Kind regards,

Mabel Aoun, MD, MPH

Academic Editor

PLOS One
---

## [Editor Report · Acceptance letter]

PONE-D-25-41837R1

PLOS One

Dear Dr. Nasser,

I'm pleased to inform you that your manuscript has been deemed suitable for publication in PLOS One. Congratulations! Your manuscript is now being handed over to our production team.

Kind regards,

on behalf of

Pr Mabel Aoun

Academic Editor

PLOS One